# Universal Adversarial Head: Practical Protection against Video Data Leakage

Jiawang Bai [1]  Bin Chen [1]  Dongxian Wu [1]  Chaoning Zhang [2]  Shu-Tao Xia [1]

## Abstract

While online video sharing becomes more popular, it also causes unconscious leakage of personal information in the video retrieval systems like deep hashing. An adversary can collect users' private information from the video database by querying similar videos. This paper focuses on bypassing the deep video hashing based retrieval to prevent information from being maliciously collected. We propose *universal adversarial head* (UAH), which crafts adversarial query videos by prepending the original videos with a sequence of adversarial frames to perturb the normal hash codes in the Hamming space. This adversarial head can be obtained just using a few videos, and mislead the retrieval system to return irrelevant videos on most natural query videos. Furthermore, to obey the principle of information protection, we expand the proposed method to a data-free paradigm to generate the UAH, without access to users' original videos. Extensive experiments demonstrate the protection effectiveness of our method under various settings.

## 1. Introduction

As an era of big data arrives, massive videos are being uploaded to the Internet, *e.g.*, YouTube, TikTok, and Twitter. For instance, the users upload more than 500 hours of video to YouTube every minute (Robertson) and more than 1 billion videos get viewed each day on TikTok (G). However, this might cause unconscious leakage of personal information by some content-based retrieval systems. For example, utilizing the advanced face video retrieval techniques (Qiao et al., 2020; Wang et al., 2020), a snoop holding a user's face video as the query can maliciously collect more personal information from online shared videos. The potential threat of the video data leakage flow is shown in Figure 1. Traditional methods for video privacy protection hide

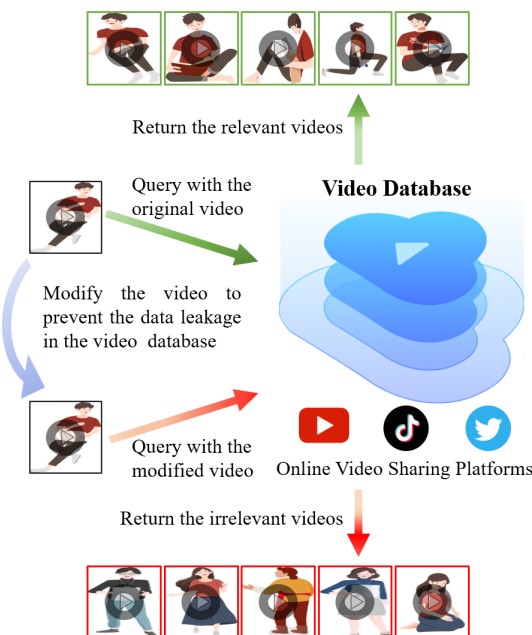

*Figure 1.* Threat of the video data leakage and the expected protection effectiveness. Relevant videos in the database can be maliciously collected by querying with the original video, while the collection utilizing the modified video fails. Best viewed in color.

the privacy information by masking (Wickramasuriya et al., 2004) or scrambling (Dufaux & Ebrahimi, 2008) the protected objects, which degrade the video quality and can not be applied to the online shared videos. Therefore, it is imperative to design a practical solution to prevent video data leakage in the retrieval system, without affecting users' experience.

Among video retrieval systems, the hashing method is promising due to its low storage cost and high search efficiency (Liong et al., 2016). It maps semantically similar videos to similar compact binary codes in the Hamming space by the hashing function. Along with the development of deep learning, deep hashing has become a mainstream video retrieval algorithm that achieves state-of-the-art retrieval performance (Li et al., 2019; Yuan et al., 2020). However, if it were exploited maliciously, the information leakage concern for the video data would be intensified. Therefore, we explore information protection on deep video hashing in this paper.

[1]Tsinghua Shenzhen International Graduate School, Tsinghua University [2]KAIST. Correspondence to: Bin Chen <cb17@tsinghua.org.cn>.

*Accepted by the ICML 2021 workshop on A Blessing in Disguise: The Prospects and Perils of Adversarial Machine Learning.* Copyright 2021 by the author(s).

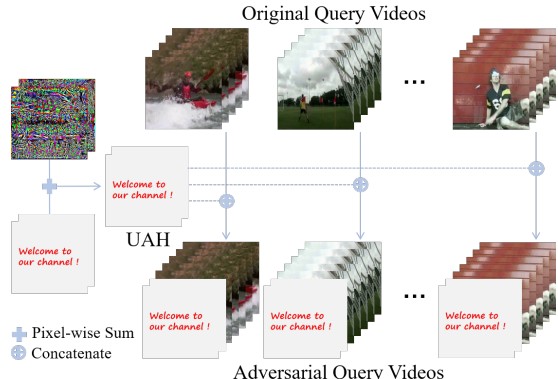

Original Query Videos

UAH

Welcome to
our channel !

⊕ Pixel-wise Sum
⊕ Concatenate

Adversarial Query Videos

*Figure 2.* Illustration of the proposed universal adversarial head.

To address the above concern, this paper studies how to bypass the deep video hashing with slight changes to the query video. In other words, the modification to the video has little impact on normal usages (*e.g.*, sharing online), but can mislead the retrieval system to return the irrelevant videos, so as to avoid malicious information collection. We demonstrate the expected effectiveness of the protection in Figure 1. Specifically, the modification method should achieve two goals: *usableness* and *stealthiness*. The usableness requires that the modification can be applied to almost all videos in the same manner without extra computation. It is especially important for practicality since benign users can adopt our technique directly on their videos. It alleviates the need for them to share their videos or generate such modifications specifically for their videos. The stealthiness requires that the modification should not affect the normal usages of the modified videos. For example, the quality of the video can not be largely decreased, so that it can be normally shared between benign users.

We accomplish the protection through a benign usage of the adversarial attack (Szegedy et al., 2014; Kurakin et al., 2017; Bai et al., 2020). The adversarial attack perturbs normal samples by intentional small perturbations and fools Deep Neural Networks (DNNs) to make incorrect predictions confidently, which has been well-studied in the machine learning security (Akhtar & Mian, 2018; Bai et al., 2021). It is usually regarded as a threat to deep learning, especially the universal attack. The universal attack calculates a universal perturbation based on only a few inputs and causes almost all inputs to be misclassified with high probability (Moosavi-Dezfooli et al., 2017; Benz et al., 2020; 2021; Zhang et al., 2020a; 2021). Inspired by this, we utilize the universal attack against deep video hashing as a modification method for information protection, which ensures usableness. However, prior works of the universal attack could not meet the stealthiness requirement, since they mainly considered the perturbations on original samples, which partly decrease the quality of the modified videos. Therefore, we propose to prepend a universal adversarial head (UAH), as shown in Figure 2, that consists of a sequence

of clean frames (dubbed clean head) with an adversarial perturbation. The content of the clean head can be specified flexibly. Although such an adversarial head is visible, the UAH does not look suspicious because many videos uploaded to the online video platforms may contain the clip of welcome, introduction, or copyright information at the beginning. Since there is no perturbation on the original frames, the UAH avoids affecting the quality of the videos. In addition, the superior performance of the UAH as demonstrated in our experiments also motivates us to apply the adversarial perturbation on the prepended head.

Once the clean head is specified, the remaining problem is to generate an imperceptible adversarial perturbation of the UAH. Utilizing some original videos, we optimize the objective, which is to enlarge the distance between the hash codes of the original videos and those of the videos with the UAH. However, in the context of information protection for the video data, we may not get access to the original videos from the users. Therefore, we further explore to generate the UAH in a more challenging and practical data-free scenario (Mopuri et al., 2017; 2018). Extensive experiments verify the effectiveness of our method under various settings.

## 2. The Proposed Method

### 2.1. Preliminaries

Let $\boldsymbol{x} \in \mathbb{R}^{M \times H \times W \times C}$ denote a given query video sampled from a distribution $p$, where $M$ is the number of frames, and $H, W$ and $C$ are the height, width and the number of channels for each frame, respectively. $f_{\boldsymbol{\theta}}(\cdot)$ is a non-linear function parameterized by $\boldsymbol{\theta}$, which transforms the input video $\boldsymbol{x}$ into a $K$-dimensional representation. A sign activation function is then used for binarizing the $K$-dimensional representation into a $K$-bit hash code $\boldsymbol{h} \in \{-1, +1\}^K$. We formulate the previous video hashing model $F(\cdot)$ as follows:

$$\boldsymbol{h} = F(\boldsymbol{x}) = \text{sign}\left(f_{\boldsymbol{\theta}}(\boldsymbol{x})\right). \tag{1}$$

The most common choice for $f_{\boldsymbol{\theta}}(\cdot)$ is the powerful feature extractor-pair with CNN+RNN (Liong et al., 2016) framework, *i.e.*, features extracted by a CNN are fed into an RNN. The hashing model $F(\cdot)$ generates a binary hash code for each video in the database and the query video. To retrieve the semantically similar videos, the similarity is measured by Hamming distance between the hash code of each video in the database and that of the query video. Then the retrieval system returns a sorted list of videos according to these computed Hamming distances.

The main focus of this paper is to seek a modification $T(\cdot)$ on the original query video $\boldsymbol{x}$ to generate an adversarial video $T(\boldsymbol{x})$, which can bypass the deep video hashing model $F(\cdot)$. For the universal adversarial attack, $T(\cdot)$ is applied to all videos sampled from a distribution $p$ in the same manner, which meets usableness requirement mentioned in

the introduction. Moreover, $T(\boldsymbol{x})$ can induce the hashing retrieval system to generate a hash code which is close to semantically irrelevant videos. Formally, we formulate it to an optimization problem as follows:

$$\max_{T(\cdot)} \mathbb{E}_{\boldsymbol{x} \sim p} \, d_H \left( F(\boldsymbol{x}), F(T(\boldsymbol{x})) \right), \qquad (2)$$

where $d_H(\cdot, \cdot)$ denotes the Hamming distance. As shown in most works on universal attack (Moosavi-Dezfooli et al., 2017; Khrulkov & Oseledets, 2018; Zhang et al., 2020b), $T(\cdot)$ is specified as directly adding the universal adversarial perturbation (UAP) to the original sample, i.e., $(\boldsymbol{x} + \boldsymbol{\delta})$, which partly decreases the quality of the video and further affects the normal usages. In addition, it is difficult to generate a universal adversarial perturbation on the original video for attacking deep video hashing as demonstrated in our experiments.

## 2.2. Universal Adversarial Head

To overcome the above challenges, we specify $T(\cdot)$ as prepending adversarial frames in front of the original query video. Thus the generation of adversarial video $\hat{\boldsymbol{x}}$ can be represented as:

$$\hat{\boldsymbol{x}} = T(\boldsymbol{x}) = [\boldsymbol{t} + \boldsymbol{\delta}, \boldsymbol{x}], \qquad (3)$$

where $\boldsymbol{t} \in \mathbb{R}^{M_t \times H \times W \times C}$ denotes the clean frames and $\boldsymbol{\delta} \in \mathbb{R}^{M_t \times H \times W \times C}$ is the adversarial perturbation on $\boldsymbol{t}$. We name $\boldsymbol{t}$ and $(\boldsymbol{t} + \boldsymbol{\delta})$ as *clean head* and *adversarial head* in this paper, respectively. $[\cdot]$ denotes the aggregation operator that combines the adversarial head and the original video into an adversarial video. The stealthiness requirement can be realized by selecting the clean head which is adaptive to the video content. The adversarial head $(\boldsymbol{t} + \boldsymbol{\delta})$ is universal, which means that it can be prepended on almost all videos from the distribution $p$ to bypass the deep video hashing.

Next we show how to generate $\boldsymbol{\delta}$ based on the adversarial head. We assume that we can get access to a video set $\boldsymbol{X} = \{\boldsymbol{x}_i\}_{i=1}^N$, where $\boldsymbol{x}_i \in \mathbb{R}^{M \times H \times W \times C}$ is sampled from a distribution $p$. To mislead the output of the deep hashing models on $\hat{\boldsymbol{x}}_i$, the perturbation $\boldsymbol{\delta}$ can be obtained by enlarging the distance between the hash code of each video $\boldsymbol{x}_i \in \boldsymbol{X}$ and that of the corresponding adversarial video $\hat{\boldsymbol{x}}_i$. Moreover, to further ensure the stealthiness, we introduce the $\ell_\infty$ restriction on $\boldsymbol{\delta}$. Then the objective function can be formulated as follows:

$$\max_{\boldsymbol{\delta}} \frac{1}{N} \sum_{i=1}^N d_H(F(\boldsymbol{x}_i), F([\boldsymbol{t} + \boldsymbol{\delta}, \boldsymbol{x}_i])), \qquad (4)$$
$$s.t. \, ||\boldsymbol{\delta}||_\infty \leq \epsilon,$$

where $\epsilon$ denotes the maximum perturbation strength.

For a pair of binary codes $\boldsymbol{h}_i$ and $\boldsymbol{h}_j$, since $d_H(\boldsymbol{h}_i, \boldsymbol{h}_j) = \frac{1}{2}(K - \boldsymbol{h}_i^\top \boldsymbol{h}_j)$, we can equivalently replace Hamming distance with inner product in the objective function. It's worth

noting that the activation function $\text{sign}(\cdot)$ in Eq. (1) is not differentiable in backpropagation. Thus, we alternatively employ $F'(\boldsymbol{x}) = \tanh(f_\theta(\boldsymbol{x}))$ to approximate $\text{sign}(\cdot)$ in the generation process. In summary, the overall optimization objective is as follows:

$$\min_{\boldsymbol{\delta}} \frac{1}{N \cdot K} \sum_{i=1}^N (F(\boldsymbol{x}_i))^\top F'([\boldsymbol{t} + \boldsymbol{\delta}, \boldsymbol{x}_i]), \qquad (5)$$
$$s.t. \, ||\boldsymbol{\delta}||_\infty \leq \epsilon.$$

## 2.3. Data-free Adversarial Generation

In the optimization problem (5), generating UAH requires access to the videos from users, which itself may be contrary to the intention of information protection. Thus, we explore the data-free scenario (Mopuri et al., 2017), in which we can provide information protection technique without access to original videos. Specifically, we are still allowed to access the target model but not the data samples from the distribution $p$, which results in a data-free variant of universal adversarial head (DF-UAH). The proposed DF-UAH consists of two major steps: 1) generate proxy query videos; 2) optimize the universal adversarial head.

The videos are necessary carriers for solving the optimization problem in Eq. (5) to obtain the UAH. Consequently, we propose to generate some proxy query videos to realize the attack. These generated videos are supposed to behave similarly like the videos sampled from the data distribution $p$, i.e., they can be mapped through the target hashing model into the Hamming space. The key idea in our method is to sample some hash codes in the Hamming space and reverse engineer the input videos that leads to the output of these hash codes. We randomly sample a hash code $\boldsymbol{h}' \in [-1, 1]^K$ to direct the proxy query video generation. We initialize the proxy query video $\boldsymbol{x}'$ as the random noise and encourage the model output to be close to $\boldsymbol{h}'$. The objective function is as follows:

$$\min_{\boldsymbol{x}'} ||F'(\boldsymbol{x}') - \boldsymbol{h}'||_2^2. \qquad (6)$$

When the loss value is optimized to approach 0, the proxy query video can be mapped into the Hamming space by the target model as the normal video. By minimizing the optimization problem in Eq. (6) with different $\boldsymbol{h}'$, we can obtain a set of proxy query videos $\boldsymbol{X}' = \{\boldsymbol{x}_i'\}_{i=1}^N$. With $\boldsymbol{X}'$, we can generate the UAH as the method in Section 2.2 by replacing $\boldsymbol{X}$ with $\boldsymbol{X}'$ in Eq. (5).

## 3. Experiments

We conduct experiments on a benchmark video dataset: UCF-101 (Soomro et al., 2012). We set the maximal length of the original video to 40. We follow the state-of-the-art video hashing methods (Gu et al., 2016; Liong et al., 2016;

*Table 1.* White-box attack results (MAP) against different architectures with various code lengths. "None" denotes the results of clean query videos. The best results are indicated in bold.

| Model | Method | 16bits | 32bits | 48bits | 64bits |
|---|---|---|---|---|---|
| $\mathcal{A}$-$\mathcal{R}$ | None | 52.60 | 51.80 | 52.76 | 49.32 |
| | UAP | 51.91 | 51.65 | 52.99 | 49.52 |
| | A²FM | 10.39 | 11.77 | 12.90 | 9.40 |
| | UAH | 6.79 | 4.21 | **5.00** | **2.85** |
| | DF-UAH | **6.75** | **3.08** | 6.07 | 4.65 |
| $\mathcal{A}$-$\mathcal{L}$ | None | 52.82 | 53.76 | 51.42 | 55.51 |
| | UAP | 52.98 | 53.79 | 52.25 | 55.28 |
| | A²FM | 21.63 | 17.15 | 20.91 | 17.93 |
| | UAH | 16.06 | **10.76** | **13.57** | **9.16** |
| | DF-UAH | **15.34** | 17.11 | 15.52 | 10.23 |
| $\mathcal{V}_{11}$-$\mathcal{R}$ | None | 54.07 | 57.85 | 55.78 | 54.76 |
| | UAP | 54.48 | 57.52 | 56.18 | 55.02 |
| | A²FM | 12.86 | 11.58 | 12.44 | 11.83 |
| | UAH | **2.43** | **2.21** | 2.58 | **2.15** |
| | DF-UAH | 2.99 | 2.26 | **2.06** | 2.23 |
| $\mathcal{V}_{11}$-$\mathcal{L}$ | None | 56.44 | 56.95 | 59.62 | 60.89 |
| | UAP | 55.29 | 56.41 | 58.53 | 59.29 |
| | A²FM | 18.07 | 15.61 | 18.68 | 17.97 |
| | UAH | **3.97** | **3.81** | **4.25** | **4.64** |
| | DF-UAH | 5.38 | 4.57 | 5.15 | 8.86 |

*Table 2.* Transfer-based black-box attack results (MAP) against different architectures. "None" denotes the results of clean query videos. The best results are indicated in bold.

| Source Model | Method | $\mathcal{A}$-$\mathcal{R}$ | $\mathcal{A}$-$\mathcal{L}$ | $\mathcal{V}_{11}$-$\mathcal{R}$ | $\mathcal{V}_{11}$-$\mathcal{L}$ |
|---|---|---|---|---|---|
| None | None | 51.8 | 53.76 | 57.85 | 56.95 |
| $\mathcal{A}$-$\mathcal{R}$ | UAP | 51.65 | 53.38 | 57.41 | 56.42 |
| | A²FM | 11.77 | 28.70 | 35.95 | 48.09 |
| | UAH | 4.21 | **20.90** | 34.29 | 42.17 |
| | DF-UAH | **3.08** | 26.23 | **33.28** | **40.00** |
| $\mathcal{A}$-$\mathcal{L}$ | UAP | 52.04 | 53.79 | 57.79 | 56.38 |
| | A²FM | 16.43 | 17.15 | 36.90 | 47.87 |
| | UAH | **10.42** | **10.76** | 35.36 | 42.98 |
| | DF-UAH | 11.75 | 17.11 | **35.13** | **39.06** |
| $\mathcal{V}_{11}$-$\mathcal{R}$ | UAP | 51.87 | 53.48 | 57.52 | 56.06 |
| | A²FM | 33.32 | 38.91 | 11.58 | 43.84 |
| | UAH | **31.49** | 34.35 | **2.21** | 26.12 |
| | DF-UAH | 33.06 | **33.82** | 2.26 | **18.07** |
| $\mathcal{V}_{11}$-$\mathcal{L}$ | UAP | 51.88 | 53.20 | 57.03 | 56.41 |
| | A²FM | 33.04 | 38.89 | 14.75 | 15.61 |
| | UAH | **31.13** | 33.97 | **5.85** | **3.81** |
| | DF-UAH | 33.00 | **33.64** | 6.73 | 4.57 |

Li et al., 2019) to choose AlexNet ($\mathcal{A}$), VGG11 ($\mathcal{V}_{11}$) and VGG16 ($\mathcal{V}_{16}$) as CNN feature extractors, and use vanilla RNN ($\mathcal{R}$) and LSTM ($\mathcal{L}$) as temporal encoders. The target model is specified by a combination of two network architectures, *e.g.*, $\mathcal{A}$-$\mathcal{R}$ denoting the hashing model with vanilla RNN following AlexNet.

We take the UAP attack (Moosavi-Dezfooli et al., 2017) as a baseline, in which the adversarial perturbation is added on the original frames. We also compare with the Appending Adversarial Frames Method (A²FM) (Chen et al., 2021), which appends a few dummy frames to a video clip and then adds adversarial perturbations only on these new frames. We adopt the standard evaluation metric mean average precision (MAP) in the video retrieval task (Gu et al., 2016) to measure the attack performance. The lower MAP value indicates a better attack performance.

For the UAH and A²FM attack, the length of the adversarial frames is 5. For all attacks, the perturbation size $\epsilon$ is 0.031 and the number of iterations is 2,000. The perturbations of UAP, A²FM, and UAH attack are computed on the set $\boldsymbol{X}$ including 100 videos randomly selected from the query set and evaluated on other 100 videos. For the DF-UAH, we generate 10 proxy videos and retain the other settings as above. We choose images with the texts of "Welcome to our channel!" as the clean frame for the UAH and A²FM attack in the following experiments (see Figure 2).

### 3.1. White-box Results

The results of the white-box attack are shown in Table 1. Under the white-box setting, all attacks are generated on the target model and used to attack this model. The UAP attack can only reduce the MAP results slightly in all cases. The MAP gaps between the query videos without and with

universal perturbations are only 0.20%. It indicates that adding the perturbation on the original frames to perform the universal attack is infeasible. In contrast, the A²FM and UAH can effectively reduce the MAP results in attacking all models with different code lengths on the two datasets. Furthermore, the results demonstrate the significant superiority of the proposed UAH compared with the A²FM. Specifically, the query videos with the UAH dramatically reduce the MAP results by 48.87% on average. Moreover, the DF-UAH attack can achieve satisfactory attack performance, even with the absence of the data.

### 3.2. Black-box Results

Table 2 shows results of the transfer-based black-box attack, *i.e.*, attacking the target model using the adversarial videos which are generated on a single source model. It shows that the UAP attack fails to generate transferable adversarial perturbations. Compared to the A²FM, our method can significantly degrade the retrieval performance of the black-box target model in all cases. We also observe that the DF-UAH is also transferable across different architectures. Moreover, an interesting phenomenon is that the transferability heavily depends on the architectures of the source and target model. More specifically, the source model whose architecture is closer to the target model achieves lower MAP results.

## 4. Conclusion

In this paper, we introduce the UAH and extend it to the data-free scenarios to protect the privacy information for the video data. We would like to emphasize that privacy protection for online shared videos deserves attention greatly. This work opens a door for applying the adversarial attack methods to the various video retrieval systems from the view of privacy protection. The more effective and stealthy methods are worth further study in the future.

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
