# OpenReview forum: "Universal Adversarial Head: Practical Protection against Video Data Leakage"
_ICML.cc/2021/Workshop/AML — ICML 2021 Workshop AML Poster_

### Official Review · Reviewer_ShK5 · 2021-06-20
**Universal Adversarial Head: Practical Protection against Video Data Leakage**

**Rating:** Accept
**Confidence:** 5

**Review:**

This paper proposes to craft the adversarial examples to apply the various video retrieval systems from the view of privacy protection. Two goals, i.e., usableness and stealthiness, are considered in this setting.  The topic of this research is interesting, while some listed conclusions seem to be further verified by implementing more experiments, such as adding diverse black-box backbones.  The author is also encouraged to study more effective black-box attack methods that can adapt to this task, because some listed results can be further improved in Table 2 while considering the importance of black-box results.

---

### Decision · Program_Chairs · 2021-06-21

**Decision:**

Accept (Poster)

**Comment:**

This paper proposed to craft the adversarial examples to apply the various video retrieval systems from the view of privacy protection. The experiments can be further improved by studying more effective black-box attacks.